

# Structural, functional and molecular dynamics analysis of *cathepsin B* gene SNPs associated with tropical calcific pancreatitis, a rare disease of tropics

Garima Singh[1], Sri Krishna jayadev Magani[1], Rinku Sharma[1], Basharat Bhat[1], Ashish Shrivastava[1], Madhusudhan Chinthakindi[2] and Ashutosh Singh[1]

[1] Department of Life Sciences, School of Natural Sciences, Shiv Nadar University, Greater Noida, Uttar Pradesh, India
[2] Department of Surgical Gastroenterology, Osmania General Hospital, Hyderabad, India

Corresponding author
Ashutosh Singh,
ashutosh.singh@snu.edu.in,
ashutosh.bio@gmail.com

## ABSTRACT

Tropical Calcific Pancreatitis (TCP) is a neglected juvenile form of chronic non-alcoholic pancreatitis. *Cathepsin B* (CTSB), a lysosomal protease involved in the cellular degradation process, has recently been studied as a potential candidate gene in the pathogenesis of TCP. According to the *Cathepsin B* hypothesis, mutated CTSB can lead to premature intracellular activation of trypsinogen, a key regulatory mechanism in pancreatitis. So far, CTSB mutations have been studied in pancreatitis and neurodegenerative disorders, but little is known about the structural and functional effect of variants in CTSB. In this study, we investigated the effect of single nucleotide variants (SNVs) specifically associated with TCP, using molecular dynamics and simulation algorithms. There were two non-synonymous variants (L26V and S53G) of CTSB, located in the propeptide region. We tried to predict the effect of these variants on structure and function using multiple algorithms: SIFT, Polyphen2, PANTHER, SDM sever, i-Mutant2.0 suite, mCSM algorithm, and Vadar. Further, using databases like miRdbSNP, PolymiRTS, and miRNASNP, two SNPs in the 3′UTR region were predicted to affect the miRNA binding sites. Structural mutated models of nsSNP mutants (L26V and S53G) were prepared by MODELLER v9.15 and evaluated using TM-Align, Verify 3D, ProSA and Ramachandran plot. The 3D mutated structures were simulated using GROMACS 5.0 to predict the impact of these SNPs on protein stability. The results from *in silico* analysis and molecular dynamics simulations suggested that these variants in the propeptide region of *Cathepsin B* could lead to structural and functional changes in the protein and thus could be pathogenic. Hence, the structural and functional analysis results have given interim conclusions that these variants can have a deleterious effect in TCP pathogenesis, either uniquely or in combination with other mutations. Thus, it could be extrapolated that *Cathepsin B* gene can be screened in samples from all TCP patients in future, to decipher the distribution of variants in patients.

## INTRODUCTION

Pancreatitis is a multifactorial, heterogeneous disease with enigmatic etiologies. It is an inflammatory condition leading to morphological changes in the pancreas, causing pain and functional abnormalities. Alcohol (*Pandol et al., 2010*), malnutrition (*Witt & Bhatia, 2008*), gallstones (*Levy et al., 2014*), familial clustering (*Kereszturi et al., 2009*) and sometimes severe infections (*Zhang et al., 2016*) have been observed to be significant causes of pancreatitis. Pancreatitis is broadly classified (*Sarner & Cotton, 1984*) as acute and chronic. Tropical Calcific Pancreatitis (TCP) (*Barman, Premalatha & Mohan, 2003*) is a juvenile form of chronic calcific non-alcoholic pancreatitis. It is a form of Idiopathic Chronic Pancreatitis (ICP), mostly reported in developing tropical countries. The phenotypic heterogeneity (*Paliwal, Bhaskar & Chandak, 2014*) includes abdominal pain, ductal dilation, large pancreatic calculi, and pancreatic atrophy. The genetic heterogeneity related to TCP is still unexplored. Fibrocalculous pancreatic diabetes (FCPD) (*Hassan et al., 2002*), a unique form of diabetes, is the unique secondary feature of TCP. TCP progresses gradually to FCPD and then at a later age, TCP patient suffers from pancreatic cancer (*Midha et al., 2010*).

The pathophysiology of the pancreas is composed of an exocrine gland which is responsible for digesting food and an endocrine gland critical for glucose homeostasis. Trypsinogen, cathepsins, serine proteases, calcium sensing receptors are some of the essential genes for pancreatic function regulation. According to the trypsin-centred theory of pancreatitis, trypsinogen, a key zymogen in pancreatic juice and a key regulator of digestion, exhibits premature activation in acinar cells of the pancreas during pancreatitis. This aberrant activation of trypsinogen leads to activation of other zymogens in the pancreas itself, thereby resulting in inflammation and autodigestion of pancreas. Although TCP is a distinct form of pancreatitis without a known cause, what remains undebated is the initial step during initiation of TCP, which is the premature activation of trypsinogen in the pancreas itself. The mortality rate in TCP is as high as 17%, and patients majorly die because of pancreatic cancer at a later stage (*Midha et al., 2010*).

Recently, we have built a database (*Singh et al., 2018*), mutTCPdb, which is a comprehensive database, giving details about the genes and variants predicted to be associated with TCP until now. Activity of trypsin inside the pancreas is the primary critical factor in pathogenesis of TCP, and all the risk genes predicted to date, are known to regulate trypsin activity like chymotrypsin C (*CTRC*), cystic fibrosis transmembrane conductance regulator (*CFTR*), serine protease inhibitor Kazal-type I (*SPINK1*) and *Cathepsin B* (*CTSB*). According to the "*Cathepsin B* hypothesis", *CTSB* plays an essential role in the premature activation of trypsinogen in the pancreas, primarily due to colocalization of *Cathepsin B* and zymogens (*Lerch & Halangk, 2006*). The precise rationale behind this colocalization is yet unknown. The reason could be aberrant trafficking mechanism of procathepsin B due to mutations in procathepsin B, or deleterious mutations in the molecules associated with the trafficking of procathepsin B in a diseased state.

A research article in 2006, described two missense mutations in CTSB (L26V and S53G), identified in TCP patients from Asian Institute of Gastroenterology, Hyderabad (India)

(*Mahurkar et al., 2006*). The minor allele frequency (MAF) of variants L26V and S53G in TCP patients were 0.46 and 0.09 respectively. Also in 2008, an article suggested that coexistence of variants in transcription factor 7-like 2 (*TCF7L2*), *SPINK1* and *CTSB* (L26V), might lead to exocrine damage in TCP and determine the onset of FCPD (*Mahurkar et al., 2008*). The analysis in this paper (*Mahurkar et al., 2008*) was performed with TCP patients and control population from Dravidian and Indo-European ethnicities. There is another article which described a missense mutation (p. Q334P) in cathepsin B gene discovered in chronic pancreatitis patients but not in TCP patients (*Xiao et al., 2017*). An article in 2014 illustrated no association of L26V mutation with TCP (*Singh, Choudhuri & Agarwal, 2014*). In this paper, statistical significance tests have indicated the lack of association of L26V mutation with TCP, but this mutation was observed in seven out of 150 TCP patients. Hence, the association of this mutation with TCP cannot be completely disregarded (*Singh, Choudhuri & Agarwal, 2014*). Although researchers have identified SNVs in CTSB gene observed in TCP patients, lacunae lie in the information about the functional effect of these SNVs in the pathogenesis of TCP.

Human *Cathepsin B* (catB, E.C 3.4.22.1) is a lysosomal cysteine protease which is involved in several cellular processes like protein degradation, extracellular matrix degradation, regulatory mechanisms, cell death, autophagy and antigen representation (*Olson & Joyce, 2015*). It belongs to papain superfamily and acts both as an endopeptidase and as an exopeptidase. *Cathepsin B* is synthesized as an inactive proenzyme (*cathepsin B*) and is activated by other proteases and by autocatalytic processes (*Pungerčar et al., 2009*). *Procathepsin B* (length of protein = 339aa) has an N-terminus propeptide of 62 amino acid length from Arg-Lys (18–79 residues). Signal sequence (1–17 residues) and post-translational glycosylation modification (*Katunuma, 2010*) targets *cathepsin B* to endosomes/lysosomes (*Ghosh, Dahms & Kornfeld, 2003*) via mannose-6-phosphate receptor pathway. Propeptide exhibits an essential role in the processing and maturation of *cathepsin B*. It acts as (a) a scaffold for catalytic domain during protein folding, (b) involved in intracellular trafficking of *cathepsin B* to lysosome after N-terminal glycosylation and phosphorylation and (c) as a high-affinity reversible inhibitor for the premature activation of zymogen. The crystal structure of procathepsin B [PDB ID: 3PBH] has a propeptide region [ArP1 to LysP62], and main chain [Leu1 to Asp254] enzyme residues (*Podobnik et al., 1997*). The main chain has two domains (R and L domains) with active site residues Cys29 and His199, located at the interdomain cleft. The propeptide siting in the active site cleft is in reverse direction to that of the substrate, thus suggesting its role as an inhibitor. The structure has an "occluding loop" [Ile105 to Pro126] which has an alternate conformation in propeptide and in mature enzyme. The occluding loop is lifted above in procathepsin B, while it is tightly packed in the active enzyme, thus exposing the active sites in *Cathepsin B*. Procathepsin B is activated by other proteases like *cathepsin D* and, also by autoactivation. The potential intermolecular cleavage site identified in *cathepsin B* is CystP42-GlyP43. At low pH, acidic residues at propeptide surface destabilize propeptide secondary structure, resulting in distortion of hydrophilic and hydrophobic interaction with mature region of protein. Subsequently, intermolecular cleavage takes place, and propeptide gets completely dissociated from mature enzyme. Thus, autoactivation is a

bimolecular process. Once CTSB gets activated, it activates trypsinogen (*Halangk et al., 2000*). Mutations affect different regions of *Cathepsin B* protein but how these variants affect the function of *Cathepsin B* is yet to be studied.

Since CTSB plays a cardinal role in premature trypsinogen activation, therefore in the present study, we decided to analyze computationally the functional and structural effect of the missense variants identified in the previous study (*Mahurkar et al., 2006*), in order to determine the clinical significance of these mutations in TCP pathogenesis. We predicted the effect of these coding variants in the propeptide region of cathepsin B, using various in silico algorithms. Also, we predicted that variants present in 3′UTR region (noncoding) in cathepsin B are associated with miRNA binding sites, and hence they could be significant. Evidential results from the structural and functional analysis of SNVs in *Cathepsin B* have implicated the potential role of these variants in the pathogenesis of tropical calcific pancreatitis. This study is the first attempt to structurally and functionally characterize the variants found in human *Cathepsin B* protein screened in TCP patients.

## MATERIALS AND METHODS

### Data curation

The single nucleotide polymorphisms (SNPs) in CTSB gene associated with TCP was extracted from an article published in 2006 (*Mahurkar et al., 2006*). In this article, researchers have done direct exome sequencing of *CTSB* gene, taking samples from 25 controls and 51 TCP patients, and further replicating the sequencing in 130 controls and 89 TCP patients from the same cohort, in order to ensure their results. In the current study, we have mapped the SNPs extracted from the literature on current human genome assembly, GRCh38.7. The mRNA accession number, NM_147782.2, and protein accession number, NP_680092.1 of gene *Cathepsin B* (CTSB), was used in our computational analysis. The current data about these SNVs were retrieved from human SNP database, dbSNPbuild150. The workflow for the computational analysis performed to decipher the significance of SNPs is depicted in Fig. 1.

### Sequence retrieval and alignment

The sequence of *Cathepsin B* (CTSB) was retrieved from UniProt database: P07858 (CATB_HUMAN). The non-synonymous variants (L26V and S53G) were manually inserted in the wild-type protein sequence for further analysis.

### Non-synonymous SNP analysis

The functional effect of mutations was predicted using the following algorithms: SIFT (Sorting Intolerant from Tolerant) (*Ng & Henikoff, 2003*), PolyPhen-2 (Polymorphism Phenotyping v2) (*Adzhubei et al., 2010*) and PANTHER (*Mi et al., 2013*). SIFT predicts whether the non-synonymous coding mutation affects protein function or not, based on sequence homology and physical properties of amino acids. SIFT calculates median conservation value for each amino acid position and thus measures the diversity of sequence. Finally, it gives the score which is the normalized probability of an amino acid change. Score of less than 0.05 are deleterious substitutions. PolyPhen-2 predicts the impact of
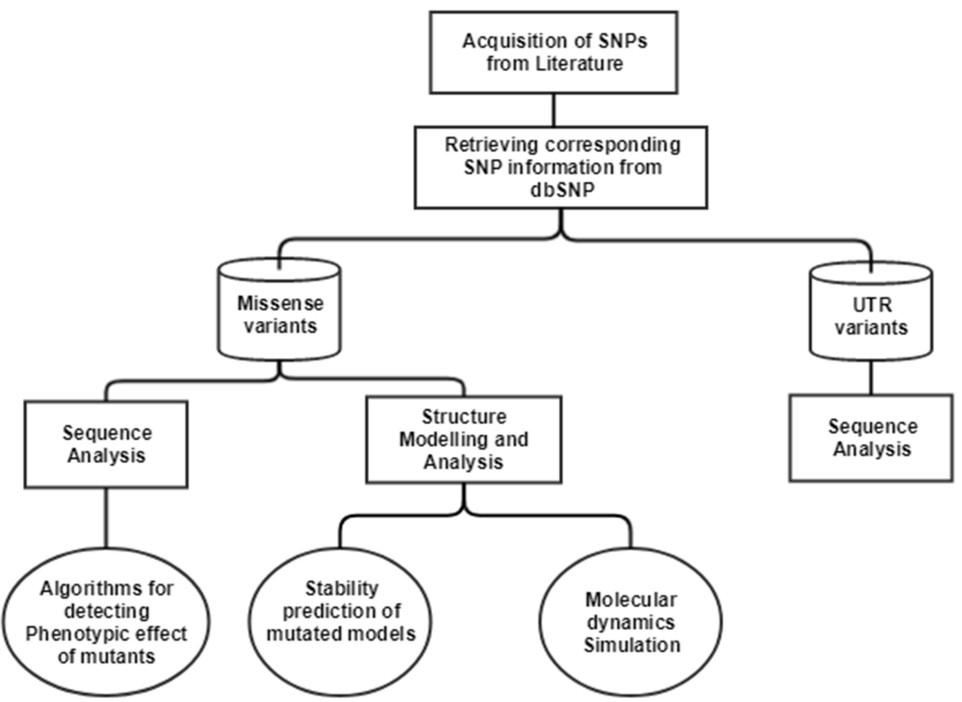

**Figure 1** Workflow to identify the potential effect of SNPs.

substitution on structure and function of a protein, after annotating the substitutions and finally building conservation profiles. The prediction algorithm of PolyPhen-2 calculates Naïve Bayes posterior probability about the damaging effect of mutation and gives prediction sensitivity scores also. PolyPhen-2 annotates the substitution as "Possibly damaging", "Probably damaging" or "Benign", based on their scores. PANTHER predicts the functional effect of coding mutation based on "evolutionary preservation" metric of a given substitution and calculates preservation time- position-specific evolutionary preservation (PESP). Longer the PESP time, the more likely that substitution will have a deleterious effect. All these softwares: SIFT, PolyPhen-2, and PANTHER, were based on evolutionary conservation-based algorithms. Another tool, ProtParam (*Bendtsen et al., 2004*) was used to calculate the hydropathicity or the GRAVY (grand average of hydropathicity) score (*Kyte & Doolittle, 1982*) of mutated procathepsin B sequences. Hydrogen bond length and the rotational angles of main chain hydrogen bonds is a significant descriptor to study the conformation and dynamics of a protein. Therefore, to calculate the altered hydrogen bonding patterns in the mutated three-dimensional procathepsin B structures, Vadar v1.8 program (*Willard et al., 2003*) was used. This program calculates the H-bond distances in the main chain, side chain and the bond angle between main chain residues.

Further, the stability of the mutated models was calculated by SDM (Site Directed Mutator) server (*Worth, Preissner & Blundell, 2011*), I-Mutant suite (*Capriotti, Fariselli & Casadio, 2005*) and mCSM (Mutation Cutoff Scanning Matrix Calculation) (*Pires, Ascher*

& *Blundell, 2014*) webservers. These algorithms calculate the difference in change in Gibbs free energy (ΔΔG). SDM server is used to calculate the difference in thermal stability of wild-type protein structure and mutated protein structure, using constrained environment specific substitution tables (ESSTs). I-Mutant suite is a support vector machine-based algorithm which predicts the protein stability upon mutation, by taking datasets from ProTherm (*Bava et al., 2004*) database. This server also calculates the change in Gibbs free energy (ΔΔG) between wild-type and mutant protein structures. mCSM server is used to predict the effect of mutations in proteins using graph-based signatures. It predicts the effect of single point mutation on protein stability by extracting data sets about various thermodynamic parameters from the ProTherm database and calculates change in Gibbs free energy (ΔΔG) between wild-type and mutant protein structures. Altogether, these three algorithms calculate ΔΔG for a protein on mutation.

The change in Gibbs free energy (ΔΔG) is as follows:

$$\Delta G = \Delta H - T \Delta S$$
$$\Delta \Delta G = \Delta G_W - \Delta G_M$$

$\Delta G$ = Change in Gibbs free energy of a system from unfavourable to favourable condition $\Delta H$ = Change in enthalpy of the system $\Delta S$ = Change in entropy of the system, T = Temperature of the system $\Delta \Delta G$ = Value of free energy stability change of a protein upon mutation $\Delta G_W$ = Change in Gibbs free energy of the wild-type protein from unfavourable to favourable conditions $\Delta G_m$ = Change in Gibbs free energy of mutant from unfavourable to favourable conditions. $\Delta \Delta G > 0$ = Increase protein stability upon mutation $\Delta \Delta G < 0$ = Decrease protein stability upon mutation.

Additionally, we have also done multiple sequence alignment (MSA) of procathepsin B protein using sequences from 8 model organisms along with *Homo sapiens* (NP_680092.1): *Mus musculus* (NP_031824.1), *Sus scrofa* (NP_001090927.1), *Macaca mulatta* (NP_001181828.1), *Rattus norvegicus* (NP_072119.2), *Ovis aries* (NP_001295516.1), *Danio rerio* (NP_998501.1), *Bos taurus* (NP_776456.1). The MSA was performed using Clustal Omega program (*Sievers et al., 2011*).

## Homology modeling

The two non-synonymous SNPs (L26V and S53G) were modelled to analyze the structural effect of variants on protein. The position specific iterated blast program (PSI-BLAST) with protein databank database (PDB) and default advanced settings, was used to find the template for homology modelling. MODELLER 9.15 (*Šali & Blundell, 1993*) was used to build mutated models of procathepsin B. The best predicted models according to the lowest value of DOPE score (Discrete Optimized Protein Energy), was used for further evaluation and analysis. The predicted 3D mutated models, L26V and S53G, were evaluated for their quality by using Verify-3D (*Eisenberg, Lüthy & Bowie, 1997*) and ProSA (Protein Structure Analysis) servers (*Wiederstein & Sippl, 2007*). Verify-3D examines the correctness of 3D-structures by comparing the 3D structure to the 1D structure. If 3D–1D score of each amino acid is ≥ 1, then the model is correct. ProSA analysis the correctness of theoretical models by calculating the $Z$-score of the input structure. ProSA considers
C-alpha atoms of protein structure and calculates $Z$-scores based on the similarity of crystal and NMR structures of the same size. If the $Z$-score for a model is negative, then it is a model with minimum or no errors. After these models pass the respective thresholds, the Ramachandran plot was evaluated (*Lovell et al., 2002*). TM-score (*Zhang & Skolnick, 2005*) and root mean square deviation (RMSD) values of mutant structures (L26V and S53G), were calculated with respect to wild-type by using TM-Align web server (*Berendsen, Van der Spoel & van Drunen, 1995*).

## Molecular dynamics simulation

The molecular dynamic simulation was performed with the Gromacs-5.0 package (*Kutzner et al., 2015*) on the native (PDB ID: 3PBH) and mutant structures (S53G and L26V). This computational investigation was done with a viewpoint to examine if these single nucleotide variants might lead to changes in surface properties or distort the protein orientation. The protein molecule was solvated in a dodecahedron box with SPC216 water molecules at 1.5 Åmarginal radiuses. The system was made neutral by adding 7 Na+(Sodium ions) because the initial charge of the system is −7. Subsequently, the molecular system was subjected to steepest distance energy minimization until reaching the criterion of 1,000 kJ/mol (the minimization is converged when maximum force is less than 1,000 kJ/mol) with OPLS-all atom force field (*Kutzner et al., 2015*). Berendson temperature coupling method (*Berendsen et al., 1984*) was used to regulate the temperature inside the box at 300 k. Isotropic pressure coupling was performed using Parinello-Rahman method (*Martoňák, Laio & Parrinello, 2003*), and the pressure of the system was maintained at 1 bar. LINCS algorithm (*Hess et al., 1997*) was used to treat bond lengths including H-bonds. Van der Waals and Coulomb interactions were truncated at 1nm, and Particle Mesh Ewald method (*Darden, York & Pedersen, 1993*) was used to compute electrostatic interaction. Finally, the simulation was performed for 35ns. The structural deviations between native and mutated structures were subjected to comparative analysis by computing RMSD (Root mean square deviation) and RMSF (Root mean square fluctuation). The trajectories were analyzed, and finally, protein compactness was studied by calculating the radius of gyration (Rg). The secondary structure analysis of wild-type and simulated mutant structures was also done using the do-dssp program of Gromacs.

## Analysis of SNPs in UTR region

5′UTR and 3′UTR regions in a gene play a crucial role in regulating gene expression at the post-translational level. UTRs regulate the exit of mRNAs from the nucleus, translation efficiency, sub-cellular localization and mRNA stability (*Mignone et al., 2002*). The effect of SNPs in UTR regions was analyzed using databases like (1). miRdSNP (*Bruno et al., 2012*), (2). PolymiRTS (*Bhattacharya, Ziebarth & Cui, 2014*) and (3). miRNASNP (*Gong et al., 2012*).

# RESULTS

## Data curation

The SNPs associated with TCP were extracted from literature and is tabulated in Table 1. The SNPs were further categorized according to their type. There were total of 23 SNPs

**Table 1** **The Single Nucleotide Polymorphisms in cathepsin B protein mined from literature (PMID: 16492714).** The SNP information is with respect to Ref Seq sequence ID: NT_077531.5 and dbSNP Build 150.

| Ref rsID/ss ID | Position | Type | CDS position (relative to CDS start) | CDS Allele change | Protein position | Residue change |
|---|---|---|---|---|---|---|
| – | Exon 1(5′UTR) | Non coding | 14,609 | C>A | – | – |
| – | Intron1(5′UTR) | Non coding | 14,520 | G>C | – | – |
| – | Intron1(5′UTR) | Non coding | 14,453 | G>A | – | – |
| rs1293311 | Intron1(5′UTR) | Non coding | 14,425 | C>A | – | – |
| rs2645415 | Intron1(5′UTR) | Non coding | 11,083 | T>C | – | – |
| – | Exon 2(5′UTR) | Non coding | 10,927 | C>G | – | – |
| rs4292649(rs12338) | Exon 3 | Non-synonymous coding(Missense) | 76 | C>G | 26 | L>V(Leu>Val) |
| rs1293293(rs1122182) | Intron 3 | Non-coding | 335 | A>T | – | – |
| – | Intron 3 | Non-coding | 394 | G>A | – | – |
| rs1293292 | Intron 3 | Non-coding | 595 | C>T | – | – |
| rs1293291 | Intron 3 | Non-coding | 663 | T>C | – | – |
| rs1803250 | Exon 4 | Non-synonymous coding(Missense) | 790* | A>G | 53 | S>G (Ser>Gly) |
| rs2272766 | Intron 5 | Non-coding | 2,609 | C>T | – | |
| rs13332 | Exon 6 | Synonymous coding | 4,383* | A>C | 140 | T>T (Thr>Thr) |
| – | Intron 6 | Non-coding | 4,451 | G>C | – | – |
| rs1736090 | Intron 6 | Non-coding | 4,735 | A>G | – | – |
| rs1692819 | Intron 7 | Non-coding | 5,516 | C>T | – | – |
| rs2294139 | Intron 7 | Non-coding | 5,522 | C>A | – | – |
| rs3215434 | Intron 7 | Non-coding(Deletion) | 5,581–5,582 | | – | – |
| – | Intron 7 | Non-coding | 5,622 | C>G | – | – |
| rs2294138 | Intron 8 | Non-coding | 5,825 | G>A | – | – |
| rs709821 | Exon 11(3′UTR) | Non-coding | 8,370* | C>G | – | – |
| rs8898 | Exon 11(3′UTR) | Non-coding | 8,422* | A>G | – | – |

(A). >sp|P07858|CATB_HUMAN Cathepsin B OS=Homo sapiens GN=CTSB PE=1 SV=3 - Native
MWQLWASLCCLLVLANARSRPSFHPLSDELVNYVKRNTTWQAGHNFYNVDMSY
LKRLCGTFLGGPKPPQRVMFTEDLKLPASFDAREQWPQCPTIKEIRDQGSCGSCWAF
GAVEAISDRICIHTNAHVSVEVSAEDLLTCCGSMCGDGCNGGYPAEAWNFWTRKGL
VSGGLYESHVGCRPYSIPPCEHHVNGSRPPCTGEGDTPKCSKICEPGYSPTYKQDKHY
GYNSYSVSNSEKDIMAEIYKNGPVEGAFSVYSDFLLYKSGVYQHVTGEMMGGHAIR
ILGWGVENGTPYWLVANSWNTDWGDNGFFKILRGQDHCGIESEVVAGIPRTDQYW
EKI

**Figure 2** **Fasta alignment of *procathepsin B protein* retrieved from Uniprot database.** (A) Fasta sequence of wild-type *procathepsin B* (NP_680092.1, Isoform 1) retrieved from Uniprot database (ID : P07858). The wild type amino acids which were mutated in TCP patients, are highlighted in red.

in CTSB gene found to be associated with TCP. The non-coding region included 20 SNPs (one deletion, six in 5′UTR, two in 3′UTR and 11 in introns). Coding region had 2 missense variants and 1 synonymous variant (Table 1).

## Sequence retrieval and alignment

The protein sequence (NP_680092.1) was retrieved from the UniProt database, and the desired variants were manually inserted in the sequence Fig. 2. The UniProt ID for the procathepsin B sequence is P07858 (CATB_HUMAN). Mutant 1 (L26V) where residue L (Leucine) is substituted with residue V (Valine). Mutant 2 (S53G) where residue S (Serine) is substituted with residue G (Glycine).

## Homology modeling

The 3D-structure of mutant (L26V and S53G) proteins was predicted after template searching by PSI-BLAST. The protein sequence (NP_680092.1) of procathepsin B was used as a query, and the resulting templates were then filtered. Finally, the X-ray crystal structure of human procathepsin B (PDB ID: 3PBH) with 2.5 Å resolution (Sequence identity: 100% and Query coverage: 93%) was used as a template for homology modeling. The DOPE scores, TM-scores, and RMSD of the predicted best models by Modeller 9.15 are shown in Table 2. The stereochemical properties of the mutated procathepsin B structures were evaluated using the Ramachandran plot from RAMPAGE. The plot defines the amino acids in favoured, allowed and outlier regions in the mutated structures as well as in the wild-type *Cathepsin B* structures (Table 3). Verify-3D did structure validation of predicted models, and it was observed that 99.05% amino acids had average 3D–1D protein score in a 21 residue sliding window ≤ 0.2 for L26V mutated model and 97.16% amino acids had average 3D–1D protein score in a 21 residue sliding window ≤ 0.2 in S53G mutated model. Additionally, the ProSA web server was also used to evaluate the quality of predicted 3D mutated models. The Z-score (by the ProSA webserver) of L26V model was −7.32 and of S53G model was −7.47, which were within the acceptable range of X-ray and NMR studies. The interaction energy analyzed by ProSA tool was negative for maximum residues in L26V, and S53G predicted models, in a sliding window of 10 and 40 respectively. Since the mutations were present in the propeptide region of procathepsin B, only the mutated propeptide structures are shown in Fig. 3 and Fig. 4.

**Table 2  Quality assessment scores after modelling protein structures.** DOPE scores after homology modelling by Modeller 9.15 of mutants (L26V and S53G) and the structure alignment scores (TM-score and RMSD) of the CTSB mutant models with wild-type, 3PBH structure.

| Predicted mutant model | DOPE Score Modeller 9.15 | TM-score TM-Align | RMSD |
|---|---|---|---|
| L26V | −34,105.02344 | 0.99941 | 0.16 |
| S53G | −34,285−52,344 | 0.99973 | 0.17 |

**Table 3  The Ramachandran plot analysis of mutated models.** The table enlists the analysis from Ramachandran plot for each of the mutated protein structures (L26V and S53G).

| Models | Favoured region | Allowed region | Outlier region |
|---|---|---|---|
| **L26V** | 92.7% | 5.1% | 2.2% |
| **S53G** | 91.7% | 6.0% | 2.0% |

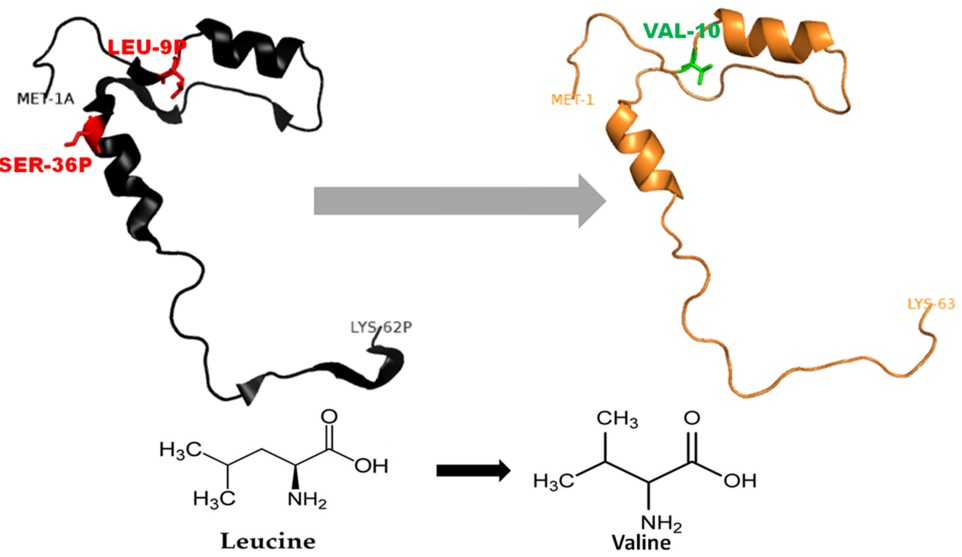

**Figure 3  Mutated (orange color) and wild-type (black color) propeptide models are shown.** The mutation, L26V is shown in sticks, which is equivalent to LEU9P-VAL10 in PDB files. Visualization and numbering is done using PyMOL tool. Note: Numbering of amino acids in wild type PDB (3PBH) file differ to that of mutated models because propeptide and peptide regions are numbered separately in the published wild-type PDB file.

## Non-synonymous SNP analysis

The functional effect of the mutations predicted by using algorithms described in the "Methods" section are tabulated in Table 4. Analysis by SIFT and PANTHER suggests that S53G and L26V mutations can have damaging effects. The GRAVY scores of wild-type WT was −0.470, and the mutants (L26V and S53G) was −0.469 for both. Thus, it could be concluded that there is no significant effect of mutations on hydropathicity of the protein. The comparative analysis of hydrogen bond lengths and the rotational angles between WT and mutants were calculated using Vadar v1.5 server, at 10 different regions of protein,
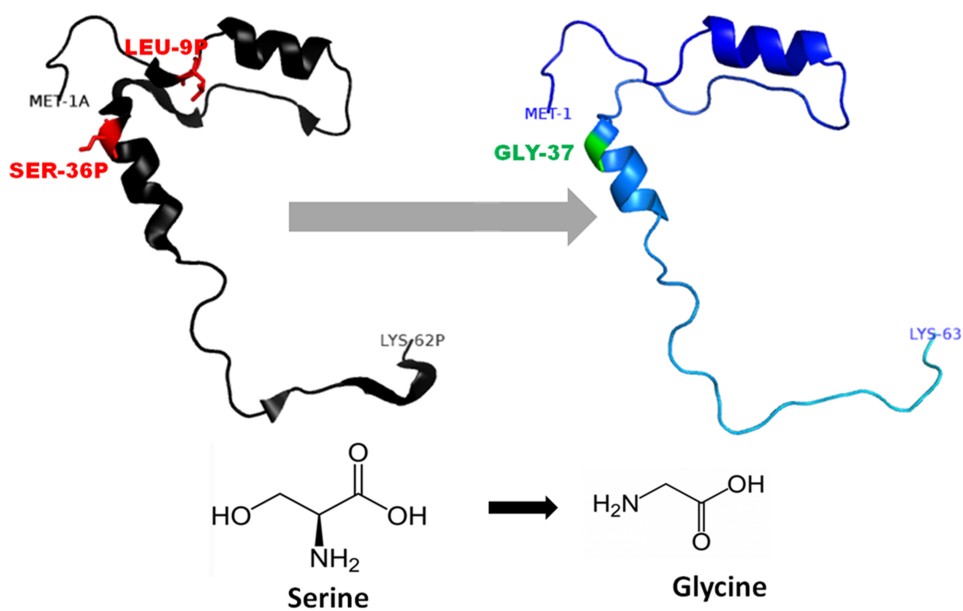

**Figure 4** **Mutated (Blue color) and wild-type (black color) propeptide models are shown.** The mutation, S53G is shown insticks as SER36-GLY37 in PDB files. Visualization and numbering is done using Py-MOL tool. Note: Numbering of amino acids in wild type PDB (3PBH) file differ to that of mutated models because propeptide and peptide regions are numbered separately in the published wild-type PDB file.

**Table 4** **Prediction of functional effect of mutations by using different algorithms.** The table enlists the scores from SIFT, Polyphen-2 and PANTHER for each mutated protein sequence: L26V and S53G.

| | | | SIFT | | Polyphen 2 | | PANTHER | |
|---|---|---|---|---|---|---|---|---|
| rsID | Allele change | AA change | Score | Prediction | Score | Prediction | Score | Prediction |
| rs4292649(rs12338) | C>G | L26V | 0 | Affect protein function | 0.01 | Benign | 1,629 | Probably damaging |
| rs1803250 | A>G | S53G | 0.02 | Affect protein function | 0.06 | Benign | 750 | Probably damaging |

playing imperative role in the functioning of procathepsin B. Remarkable differences in the H-bond lengths and Bond angles between WT and mutants (S53G and L26V) were observed as represented in Fig. 5. Hence, it could be interpreted that these mutations alter the binding between residues in mutated structures.,

The change in Gibbs free energy ($\Delta \Delta G$) of the mutated structures calculated by SDM server, I-Mutant 2.0, and mCSM webservers indicateddestabilization of mutated proteins (Table 5). Additionally, the results of MSA revealed that leucine at the 26th position and Serine at the 53rd position are conserved residues. Extrapolating these results indicate that any alteration in conserved residues will affect the structure and function of the protein (Fig. 6). Altogether, L26V and S53G mutations were predicted to have a deleterious effect on the structure and function of the protein, through non-synonymous SNP analysis algorithms.
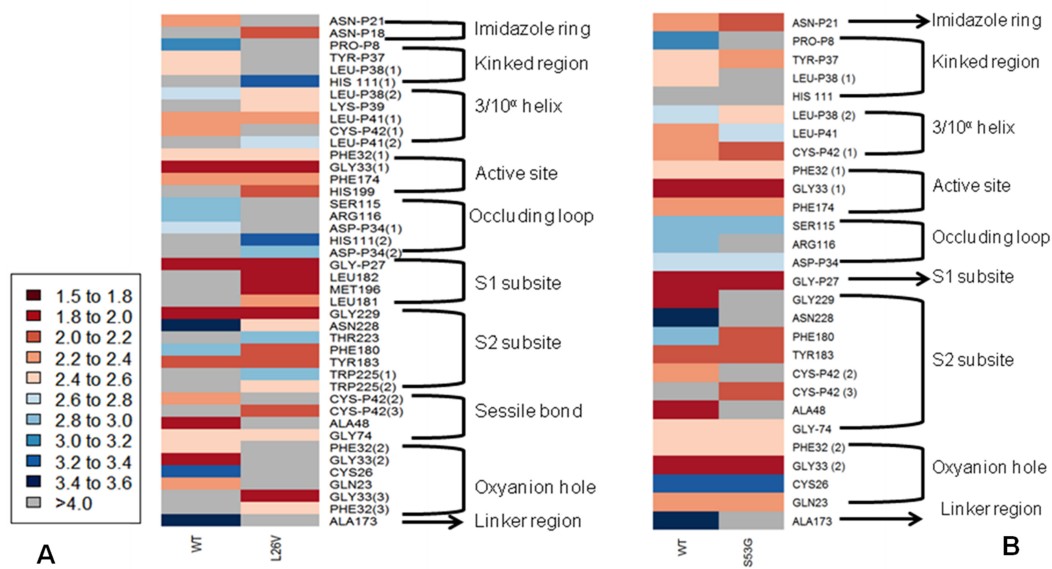

**Figure 5** **The comparative analysis of H-bond length between wild-type (WT)** *procathepsin B protein* **(PDB ID: 3PBH) and.** (A) Mutated structure (L26V) having a mutation at the 26th amino acid (Leucine to Valine) in the propeptide region (B) Mutated structure (S53G) having a mutation at the 53rd position (Serine to Glycine) in the propeptide region. The colour key ranges from 1.5 Å to 3.5 Å with red as strong H-bonding and blue as weak H-bonding. Wide spaces indicate the absence of H-bonds at that position.

**Table 5** **Prediction of protein (procathepsin B) stability upon mutation.** The table enlists the change in Gibbs free energy ($\Delta\Delta G$) in kcal/mol. $\Delta\Delta G > 0$ indicates stabilization while $\Delta\Delta G < 0$ indicates destabilization.

| Algorithm | S53G | L26V | Effect |
|---|---|---|---|
| SDM | −.174 | −0.94 | Destabilizing |
| I-Mutant 2.0 | −1.48 | −1.93 | Destabilizing |
| mCSM | −1.063 | −1.638 | Destabilizing |

## Analysis of SNPs in UTR region

UTRs play an essential role in mRNA processing during post-translational mechanism. Hence, the SNPs in the UTR region can significantly affect the functionality of UTRs, provided they affect the miRNA binding sites. The 3′UTR region is essential for microRNA (miRNA) binding which can lead to degradation or transcriptional suppression of mRNA and thus, can further affect the downstream processing. The databases miRdbSNP, PolymiRTS, miRNASNP, were used to predict the significance of SNPs in the 3′UTR region (Table 6). The two SNPs present in 3′UTR region, rs709821 and rs8898, were predicted to be present in miRNA binding sites and therefore are significant. The SNP, rs8898 was predicted to create a new miRNA site, while rs709821 disrupt a non-conserved miRNA site as predicted by PolymiRTS database. The two miRNAs targeting CTSB gene having both non-coding variants, hsa-miR-96 and hsa-miR-1271, are pancreas specific miRNAs, deciphered from the miRNet database (*Fan et al., 2016*).

**Figure 6** **Multiple sequence allignment.** (A) MSA of wild type procathepsin B protein (NP_680092.1). (B) MSA of mutated protein sequence with mutation S53G. (C) MSA of mutated protein sequence with mutation L26V.

### Molecular dynamics simulations

The comparative analysis of trajectories by calculating RMSD, RMSF and Radius of gyration after MD simulation of 35 ns, for both native and mutants, was performed. Interestingly, it was observed from RMSD of backbone residues, that both mutated structures (S53G and L26V) were conspicuously deviated from the native structure (PDB ID: 3PBH). To infer effect of mutations on the dynamic behaviour of each residue, RMSF of C $\alpha$-atoms was calculated. There was fluctuation observed in mutated protein structures as compared to the native structure. The protein compactness was determined by the radius of gyration (Rg). It was observed that Rg of mutated structures were distinctly fluctuated as compared to native structure, throughout the simulation (Fig. 7). The secondary structure analysis of wild-type and simulated mutant structures, by do-dssp program, implicated that mutations had caused deviation to the protein (Fig. 8).

## DISCUSSION

Tropical Calcific Pancreatitis (TCP) has distinct morphological characteristics with undefined etiology. The propeptide region of procathepsin B (Arg1 -Lys 62) i.e., the N-terminal part inhibits the activity of *Cathepsin B* in the pancreas, thereby regulating its premature activation, also act as a scaffold for protein folding and as a chaperone for

**Table 6 The SNPs in 3′UTR region of CTSB protein.** The table enlists the predicted miRNAs targeting the CTSB gene sequence having 3′UTR SNVs.

| rsID | Region | Allele change | miRdSNP | PolymiRTS | miRNASNP |
|------|--------|---------------|---------|-----------|----------|
| rs709821 | UTR-3 | C>G | hsa-miR-186 | – | – |
| | | | hsa-miR-339-5p | – | – |
| | | | hsa-miR-7 | – | – |
| | | | hsa-miR-214 | – | – |
| | | | hsa-miR-431 | – | – |
| | | | hsa-miR-186 | – | – |
| | | | hsa-miR-320a | – | – |
| | | | hsa-miR-320d | – | – |
| | | | hsa-miR-320c | – | – |
| | | | hsa-miR-320b | – | – |
| | | | hsa-miR-96 | – | – |
| | | | hsa-miR-1271 | – | – |
| rs8898 | UTR-3 | A>G | hsa-miR-339-5p | hsa-miR-10a-5p | hsa-miR-10a-5p |
| | | | hsa-miR-186 | hsa-miR-10b-5p | hsa-miR-10b-5p |
| | | | hsa-miR-7 | hsa-miR-339-5p | hsa-miR-339-5p |
| | | | hsa-miR-214 | hsa-miR-4421 | |
| | | | hsa-miR-431 | hsa-miR-5699-3p | |
| | | | hsa-miR-186 | hsa-miR-6747-3p | |
| | | | hsa-miR-320a | hsa-miR-6752-3p | |
| | | | hsa-miR-320d | | |
| | | | hsa-miR-320c | | |
| | | | hsa-miR-320b | | |
| | | | hsa-miR-96 | | |
| | | | hsa-miR-1271 | | |

endosome/lysosomal trafficking. L26V and S53G are the two missense variants observed in the propeptide region of *Cathepsin B* protein in TCP patients. The *in-silico* SNP analysis of the mutated protein sequences, resulted in alteration of secondary structure, thereby predicting an adverse folding effect on the protein. The phenotypic effect of the mutations was calculated using sequence analysis algorithms, and it was observed that at least two algorithms indicated a deleterious effect of these mutations on protein functionality. The free energy ($\Delta\Delta G$) calculations of mutated proteins structures, by various algorithms, indicated that mutations are destabilizing. Further, the comparative analysis of H-bond distances between mutated and native 3D-structure of procathepsin B provided a unique information about the structural characteristics of motifs around main chain H-bonds which are altered in mutant protein structures, thereby affecting the function of the protein (*Penner et al., 2014*). Additionally, MD simulation of mutated and native protein structures indicated that the mutations distinctly deviate the structural conformation of procathepsin B, thereby having a deleterious effect on downstream signalling mechanism. Thus, the structural and functional analysis of mutated procathepsin B predicts the significance of these mutations in the propeptide region of *Cathepsin B*. Hence, we could extrapolate

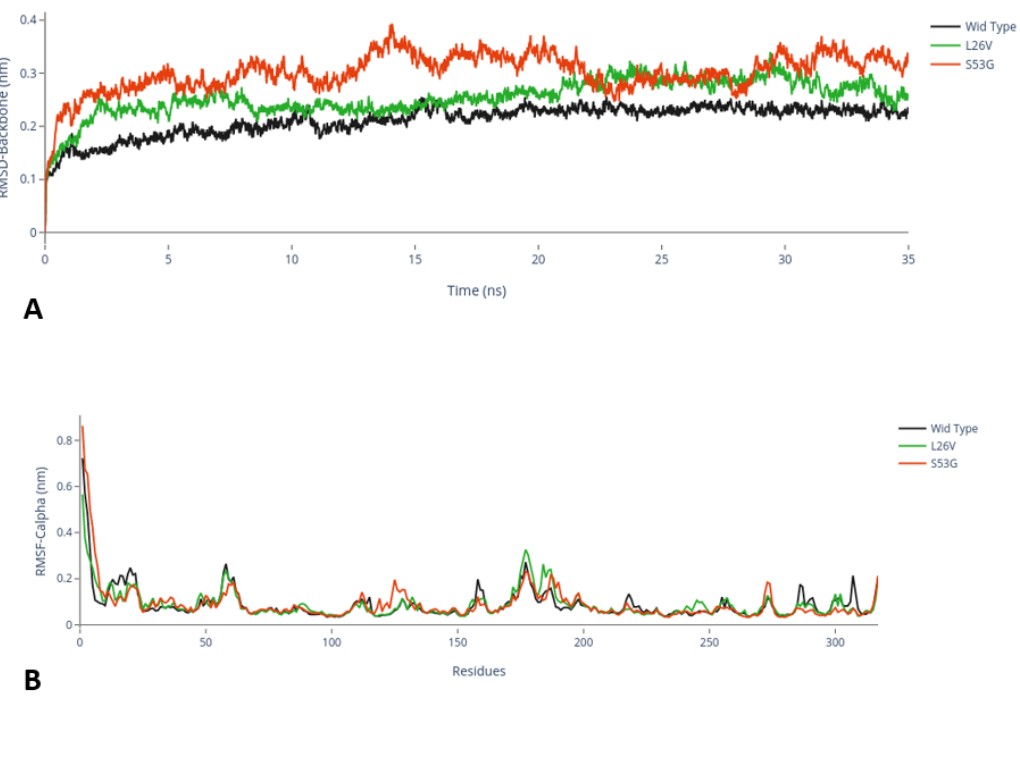

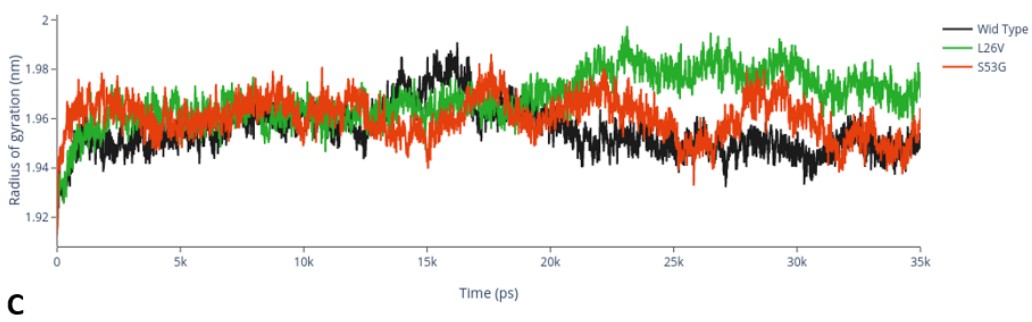

**Figure 7  MD Simulation.** (A) Root mean square deviation (RMSD)-Backbone: Comparative analysis of RMSD-backbone between Wild-type and both mutated structures. (B) Root mean square Fluctuation (RMSF)-C-alpha: Comparative analysis RMSF between mutated (s53g and l26v) and native protein structures. (C) Radius of gyration (Rg): Comparative analysis of Rg between wild-type protein (s53g, l26v) and native protein structures.

from these results *(in silico* analysis of the mutated structures and sequences) that both mutations (L26V and S53G) have a deleterious effect on structure and function of protein. These results will provide a lead towards designing the experimental research strategy on the mutations involved in the pathogenesis of TCP to understand the disease etiopathogenesis.

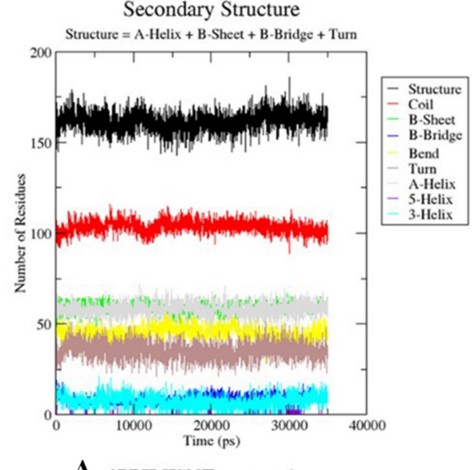

**Figure 8 Secondary structure analysis.** The graphs depicts the secondary structure analysis using do-dssp program of (A) Wild type protein structure—3PBH (B). Mutated structure 1: L26V and (C) mutated protein structure 2: S53G.

## CONCLUSION

The dearth of information about the etiopathogenesis of tropical calcific pancreatitis was the driving force for this study. The literature only has the information about the SNVs in the *cathepsin B* gene associated with TCP and lacks the crucial theories about the relative effects of these SNVs in the pathogenesis of TCP. In this study, we predicted the structural and functional effect of *cathepsin B* SNVs which were identified in TCP patients in previous studies. The predicted deleterious effect of these SNVs is a lead towards developing

biomarkers and therapeutics for TCP. Further studies in this direction will help in defining the pathophysiology of TCP, which is still a conundrum.

### Funding

This work was supported by Shiv Nadar University, Greater Noida, India. The funders had no role in study design, data collection and analysis, decision to publish, or preparation of the manuscript.

### Grant Disclosures

The following grant information was disclosed by the authors:
Shiv Nadar University, Greater Noida, India.

### Competing Interests

The authors declare there are no competing interests.

### Author Contributions

- Garima Singh conceived and designed the experiments, performed the experiments, analyzed the data, contributed reagents/materials/analysis tools, prepared figures and/or tables, authored or reviewed drafts of the paper.
- Sri Krishna jayadev Magani conceived and designed the experiments, contributed reagents/materials/analysis tools, authored or reviewed drafts of the paper.
- Rinku Sharma performed the experiments, contributed reagents/materials/analysis tools, prepared figures and/or tables.
- Basharat Bhat performed the experiments, contributed reagents/materials/analysis tools.
- Ashish Shrivastava performed the experiments.
- Madhusudhan Chinthakindi conceived and designed the experiments, contributed reagents/materials/analysis tools.
- Ashutosh Singh conceived and designed the experiments, contributed reagents/materials/analysis tools, authored or reviewed drafts of the paper, approved the final draft, edited manuscript, overall management of the project.

### Data Availability

The raw data used in this study are available at NCBI via PMC accession number GRCh38.7, mRNA accession number NM_147782.2, and protein accession number NP_680092.1. The SNV data was gathered from literature (*Mahurkar et al., 2006*) and is available in Table 1.

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
