# Peer review of "Structural, functional and molecular dynamics analysis of cathepsin B gene SNPs associated with tropical calcific pancreatitis, a rare disease of tropics"

_PeerJ, doi:10.7717/peerj.7425_

## Round 0.1 · original submission · Major Revisions

Please address all the critical issues raised by both reviewers and amend your manuscript accordingly.

Reviewer 1 ·

Basic reporting

The English language must be improved to ensure that an international audience can clearly understand your text. To highlight- a few examples in the abstract where language could be improved: - there is no word "destable" in the English language. Consider using "less stable" or unstable etc. In the next sentence "Results from MD-simulation carried for both mutated structures have concluded that mutations lead to structural deviation in protein..." - results are

"..these variants can have deleterious" - the word deleterious should be preceded by an article (a or the).

"Thus cathepsin B gene should be screen in samples" - "Thus cathepsin B gene should be screened in samples...."

"variants can have deleterious effect" - "variants can have a deleterious effect"

"...distribution of variations in patient population" - "distribution of variations in patient the population"

The manuscript is full of such errors, I can't point all of them.

Experimental design

1.Kindly explain why PSI-BLAST (and not BLASTp) has been used to query PDB sequences? How much % identify in the alignment has been used for homology modeling?
2. It will be useful to show contour maps of the free energies as a function of the backbone angles ψ and φ for residues, AND Energies for binding can be calculated. see: - https://www.ncbi.nlm.nih.gov/pmc/articles/PMC4926353/, https://www.ncbi.nlm.nih.gov/pmc/articles/PMC3607329/:
3. Please include results to show secondary structure using do-dssp.
4. MD simulation time should be increased to check longer time-dependent changes.
5. Combined effect of mutation on protein function can also be observed in order to understand their effects.
6. MSA from known model organism procathepsin B proteins can be performed with both wild and mutant.
7. In the section "Materials and methods": - the section "Non-synonymous SNP Analysis" should come before "Homology modeling"
8. Abbreviation should be defined at it's first occurrence, and one time only. Abbreviation for SIFT [Sorting Intolerant From Tolerant] is defined in the line numbers 133 as well as 180- similarly check other abbreviations too.

Validity of the findings

The findings are novel and interesting, however, minor additional analysis, it's integration with other findings combined with language corrections will enhance its impact on readers.

Reviewer 2 ·

Basic reporting

The article would benefit from being checked by a native English speaker. There are many small mistakes throughout the article (eg. missing "the" lines 66, 86, 99).

The introduction contains methods, which are then repeated in the methods section (lines 123-143). Additionally, it contains a large amount of text on the protein structure, I am not sure that this detailed information is relevant for the article's purpose. Many references for the non-synonymous SNP methods section are missing (although in the Intro), and the reference style appears to be inconsistent (e.g. line 202). References are missing for the miR databases in line 371.

Detail on the prevalence of TCP, and how it is increasing mortality rate in tropical countries (line 73) would be beneficial.

Figures and tables: I do not see the need for Fig 2, the numbers are contained in the text and are also in Table 1. Figure 3 could be shortened to only have the sequence for the WT protein, with the two mutations highlighted. Figure 4 and 5 refer to "sticks" - is this the proper terminology for these diagrams? Line 337 refers to Fig 5, is this correct?

Note: Table 1 appears to have the header repeated.

Experimental design

The premise of the article relies on SNPs found to be associated with TCP in a 2006 study. No details of the size of this study were given (here some background information on prevalence of TCP in the general population would also potentially justify using a small study), are there any more recent studies available that could also be used? Is the association with those SNPs replicated in later studies? Which human genome build was used? hg18/19/37?

Are there other SNPs in the CTSB gene? How do they compare cf function/location with respect to those that were found to be associated with TCP?

miRs are often tissue specific. What are the targets of the miRs listed in table 6, and are they known to be expressed in any TCP relevant tissues? There is the potential to analyse this data much more than has been done, noting that it is all in silico prediction only.

The Methods section lists a large number of programs that were used, but some details are missing. For instance, "After these models pass the respective thresholds" (line 173) however no thresholds are given in this paragraph.

Did the authors perform any downstream analysis on the output of any of the programs used?

Validity of the findings

No comment

External reviews were received for this submission. These reviews were used by the Editor when they made their decision, and can be downloaded below.

---

## Round 0.2 · Minor Revisions

Please address the remaining critiques of the reviewer #2.

Reviewer 2 ·

Basic reporting

The article stills need to be checked completely for English and grammar. For example:

line 47: missing full stop
line 50: This lead to activation - "This leads to activation"
line 51: Double full stop; Sentence beginning with "TCP" does not make sense
line 53: Double full stop
line 54: patients majorly - "the majority of patients"

There are numerous other such examples still present in the article.

In the introduction, the authors should group all statements about what they have done into one paragraph, currently its on lines 74-80 and 110-115.

Figures and tables: I still do not see the point of Figure 2 - it is not a complicated figure, and the numbers are in the text. It doesn't add to the paper and could easily be removed.

Experimental design

The additional details regarding the original 2006 study improve the manuscript. However, I think the authors are misleading in their use of language to describe this study. The 2006 paper investigated 23 SNPs in the CTSB gene, but found only 1 SNP to be significantly associated with TCP after adjusting for multiple corrections. And indeed, only attempted to validate 4 of the 23 SNPs. Why then the focus on these 23 SNPs, given that the majority of them are NOT associated with TCP?

The authors state that they couldn't find any other CTSB SNP to be associated with TCP - how did they come to this conclusion? Did they do any association testing themselves, or just a literature search?

My original question of how do these other non-associated CTSB SNPs compare to the TCP-associated SNPs remains unaddressed. This is important, especially now that it appears that many of the TCP-associated SNPs (as defined by the authors) are actually NOT associated with TCP.

Validity of the findings

While the findings might be robust, the premise on which the SNPs were chosen in the first place is not (see my comments above).

External reviews were received for this submission. These reviews were used by the Editor when they made their decision, and can be downloaded below.

---

## Round 0.3 · Minor Revisions

Although the majority of the critiques of the reviewers were adequately addressed, the manuscript still requires some revision.
1) You stated that there is a lack of information on other SNPs of cathepsin, after a few minutes in Clarivate's Web of Science a paper was found (10.1016/j.jpeds.2017.08.063) describing an extra cathepsin mutation (Q334P) possibly involved in pancreatitis, though not tropical calcific pancreatitis. This point should be discussed.
2) Furthermore, you did not mention another study, 10.1155/2014/763195, which failed to find any increased prevalence of the L26V mutation in pancreatitis patients. This point should be discussed too.
3) The whole homology modeling section is mostly uninformative due to the magnitude of the change amounting to 0.5% of the sequence length. When one attempts to model something from a low-identity template, all those quality scores are good enough to understand if the model makes sense. Starting from a 99.5% identity experimental template, those tools will obviously return quality scores mostly indistinguishable from those of the experimental structure.
4) Finally, Panel B in Figure 7 has two black and two green traces up to residue 55

External reviews were received for this submission. These reviews were used by the Editor when they made their decision, and can be downloaded below.

---

## Round 0.4 · accepted · Accept

In my view, all the remaining critiques were addressed and the manuscript was adequately amended.

External reviews were received for this submission. These reviews were used by the Editor when they made their decision, and can be downloaded below.